# LEARNING INVARIANT GRAPH REPRESENTATIONS VIA VIRTUAL ENVIRONMENT INFERENCE

## ABSTRACT

Graph invariant learning aims to learn invariant graph representations across different environments, which achieves great success in tackling Out-of-Distribution (OOD) generalization in graph-related tasks. As environments on graphs are usually expensive to obtain, most graph invariant learning methods heavily rely on inferring the underlying environments to learn the environment-wise invariant graph representations. Actually, inferring the underlying environments is extremely challenging, due to the high heterogeneity of the graph environments and the unknown number of underlying environments. In this paper, we solve the OOD graph generalization task from a class-wise perspective, enabling us to generate more reliable virtual environments for effective graph invariant learning. This is motivated by the observation that class-wise spurious features are more likely shared by different classes despite high environment heterogeneity. To this end, we introduce a novel framework, named **C**lass-wise invariant risk minimization via **V**irtual **E**nvironment **I**nference (C-VEI), which aims to discard class-wise spurious correlations and preserve class-wise invariance. Specifically, to infer the class-wise virtual environments, C-VEI introduces a contrastive strategy on the latent space, which i) pulls samples from the same class but dissimilar graph representations together and ii) pushes samples from different classes but similar graph representations away. In addition, we design a class-wise invariant risk minimization to preserve class-wise invariance, We conduct extensive experiments on several graph OOD benchmarks and demonstrate the consistent superiority of our C-VEI across all settings and metrics. The source code will be made publicly available.

## 1 INTRODUCTION

Graph representation learning with graph neural networks (GNNs) has gained great success and has a wide range of realistic applications, *e.g.,* drug discovery (Gaudelet et al., 2021) and recommender systems (Ahmad & Lin, 1976). Despite the success, the existing graph representation learning methods heavily rely on the independent and identically distributed assumption, *i.e.,* the testing and training graph data are independently drawn from the same distribution. However, in reality, such an assumption is difficult to satisfy as the incoming graph data can easily be affected by some underlying *environmental factors* (Ji et al., 2023; Gui et al., 2022). Inspired by causal invariance principle (Peters et al., 2016) and Invariant Risk Minimization (Arjovsky et al., 2019) (IRM) and graph invariant learning (Wu et al., 2022; Li et al., 2022b; Chen et al., 2022b) is proposed to learn invariant graph representations with respect to an invariant subgraph across different environments, such that the predictions made based on the invariant subgraphs can generalize to unknown distribution.

Due to the abstraction of graph data, the environment partitions on graphs are usually unavailable. Hence, most existing graph invariant learning approaches focus on inferring the underlying environment labels (Wu et al., 2022; Li et al., 2022b; Yang et al., 2022). However, in many realistic applications such as drug affinity predictions (Ji et al., 2023), the heterogeneity of the training environments is often high: (a) the number of data from each environment can be too small to cover all of the classes; (b) the number of the underlying environments can be too large to infer reliably. Without strong assumptions or more inductive bias, it is extremely challenging to uncover the *full underlying environment labels*. Take the most representative graph invariant learning approach GIL (Li et al., 2022b) as an example, GIL first partitions the input graphs as the invariant and variant subgraphs, and infers the environments via clustering on the variant subgraphs. GIL can fail catastrophically even in

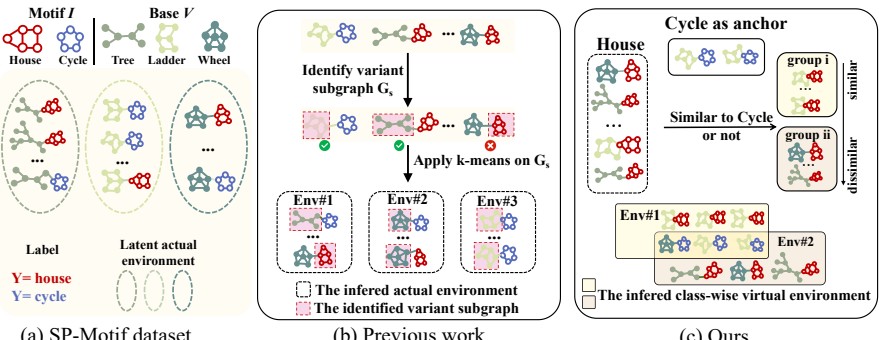

(a) SP-Motif dataset       (b) Previous work       (c) Ours

Figure 1: (a). SP-Motif dataset is a synthetic dataset where each graph consists of one variant subgraph (*i.e.,* base, denoted by $V$) and invariant subgraph (*i.e.,* motif, denoted by $I$); the ground truth labels $Y$ only depends on invariant subgraph $I$ but variant subgraph V spuriously correlates $Y$. (b). GIL (Li et al., 2022b) applies K-means to infer the underlying environments. However, due to the high environment heterogeneity, the learned environments by GIL (Li et al., 2022b) are inaccurate (Env#1) or only contain samples from the same class (Env#3). (c). Our C-VEI creates two specific environments for each class to remove its spurious correlation. With the assistance of labels, the inferred class-wise virtual environments are more reliable and interpretable.

a toy setting (see Fig. 1(a)), where "*House*" spuriously correlates "*Tree*" base and "*Cycle*" spuriously correlates "*Ladder*". As shown in Fig. 1(b), *without sufficient data*, the subgraph partitioning can easily be misled by spurious subgraphs, which further exacerbates the biases in the inferred environments (Env#1 and Env#3). Moreover, *without knowing the number of the underlying environments*, the inferred environments can be too diverse to cover all classes or too dense to cover all the spurious patterns in each environment. Therefore, it raises a challenging question:

*How can we generate reliable environments for graph invariant learning under heterogeneous environments?*

To tackle the challenge, we propose a novel graph invariant learning framework, termed Class-wise invariant risk minimization via Virtual Environment Inference (C-VEI) for graph OOD generalization. Instead of inferring the *underlying environments*, C-VEI creates *virtual environments* to discard spurious subgraphs from a class-wise perspective. Our key observation is that **spurious features are more likely shared by different classes despite high environment heterogeneity**. Thus, it is easier to identify the spurious subgraphs by contrasting samples that are similar but from different classes.

We thus are motivated to propose to use each class as an anchor to split the samples of the rest of the classes into two groups: similar to the anchor or not. As a result, for $C$ classes, we will have a total of $2C$ approximately virtual environments. Finally, we apply the IRM objective function based on contrastive learning on the inferred virtual environments to remove spurious correlation. As shown in Fig. 1(c), when "*Cycle*" as the anchor, we first split other class graphs into two groups: (i) similar to "*Cycle*" and (ii) others. Considering that the "*Cycle*" class spuriously correlates with the "*Ladder*" base, the group (i) more likely contains graphs with the "*Ladder*" base. In Env#1, the model is required to identify "*Cycle*" graphs from most graphs with "*Ladder*", so that it is easy to remove the spurious correlations between the "*Cycle*" and the "*Ladder*".

C-VEI consists of four modules: a Rationale Generator (RG), a Class-wise Virtual Environment inference module (CVE), a GNN feature encoder, and a classifier. Specifically, RG learns to split the input graph into rationale subgraphs and variant subgraphs, which are respectively encoded by the feature encoder into representation. Then, CVE constructs two virtual environments for each class according to the similarity of rationale subgraph representations. Finally, we apply a class-wise IRM objective to perform contrastive learning under the constructed environments. To sum up, our main contributions are as follows:

- We identify several challenges prohibiting the environment inference for graph invariant learning.
- We propose a unified class-wise graph invariant learning framework, C-VEI, which aims to heuristically create class-wise virtual environments to disregard the variant spurious correlations and learn invariant representations for graph OOD generalization. With the assistance of labels, the inferred class-wise virtual environments are interpretable and reliable.
- We conduct extensive experiments on both synthetic and real-world graph OOD generalization datasets, and show that our C-VEI achieves new state-of-the-art results.

## 2 RELATED WORKS

**Graph Out-of-Distribution Generalization.** Graph Out-of-distribution (OOD) generalization (Li et al., 2022a; Chen et al., 2023; Liu et al., 2023; Zhu et al., 2023; Li et al., 2023; Gui et al., 2023) aims to achieve satisfactory generalization performance under distribution shifts, which facilitates graph machine learning model deployments in real-world scenarios. As a promising learning strategy to achieve graph OOD generalization, graph invariant learning aims to exploit the invariant relationships between invariant features and labels across different environments to make OOD generalizable predictions (Li et al., 2022a; Creager et al., 2021; Wang et al., 2022; Chen et al., 2022a). Due to the abstraction of graph data, in reality, graph data usually comes from a mixture of latent environments without accurate environment labels (Chen et al., 2022b). Hence, most existing graph invariant learning approaches rely on inferring the underlying environment labels (Wu et al., 2022; Li et al., 2022b; Yang et al., 2022). However, in many realistic applications such as drug affinity predictions (Ji et al., 2023), the heterogeneity of the training environments is often high: i) The number of underlying environments may be too large to be reliably inferred. ii) The number of data samples from each environment may be insufficient to represent all classes. For example, DrugOOD (Ji et al., 2023), a systematic graph OOD benchmark for AI-aided drug discovery, has a total of $186, 875$ scaffold domains but only $568, 556$ samples. Considering the high heterogeneity environments, without strong assumptions (Wu et al., 2022; Chen et al., 2022b; Li et al., 2022b; Yang et al., 2022; Chen et al., 2023) or more inductive bias, it is extremely challenging to uncover the *full underlying environment labels*. The falsely inferred environments can further exacerbate the bias the invariant learning and lead to degenerated OOD generalization. Instead of inferring the factual environments, we propose to generate class-wise virtual environments. With the assistance of class labels, it is relatively easier to identify spurious subgraphs in the virtual environments.

**GNN Explainability.** GNN explainability aims to find an explanation in the form of input features with the maximum influence over the prediction. These explanations denote rationale subgraphs, which can be a set of either node features or a substructure (set of nodes/edges) or both (Ying et al., 2019; Yuan et al., 2020). Recently there are several inherently interpretable models (Miao et al., 2022; Yu et al., 2021; Wu et al., 2022) build relationships between the explainability and OOD generalizations of GNNs. Inspired by the information bottleneck (IB) principle (Tishby et al., 2000), GIB (Yu et al., 2021) and GSAT (Miao et al., 2022) both aim to identify a rationale subgraph to interpret GNNs by optimizing an IB objective. Moreover, DIR (Wu et al., 2022) aims to provide robust explanations under distribution shifts from a causality perspective. Following Chen et al. (2022b); Li et al. (2022b), we also select the state-of-the-art GNN explanation methods (Yu et al., 2021; Miao et al., 2022; Ranjan et al., 2020; Wu et al., 2022) as baselines.

## 3 PROBLEM FORMULATION

In this work, we focus on OOD generalization in graph classification. Specifically, we are given a set of graph datasets $\mathcal{D} = \{\mathcal{D}^e\}_{e \in \mathcal{E}_{\text{all}}}$ collected from multiple environments $\mathcal{E}_{\text{all}}$. Samples $(G_i^e, Y_i^e) \in \mathcal{D}^e$ from the same environment are considered as drawn independently from an identical distribution $\mathbb{P}^e$. The environment labels for graphs are unobserved since it is expensive to collect environment labels for most real scenarios. The goal of OOD generalization on graphs is to train a GNN $f$ with training dataset $\{\mathcal{D}^e\}_{e \in \mathcal{E}_{\text{tr}} \subseteq \mathcal{E}_{\text{all}}}$, which generalizes well to all (unseen) environments:

$$f^*(\cdot) = \arg\min_{f} \sup_{e \in \mathcal{E}_{\text{all}}} \mathcal{R}(f|e), \tag{1}$$

where $\mathcal{R}(f|e) = \mathbb{E}_{G,Y}^e[\mathcal{L}(f(G), Y)]$ is the risk of the predictor $f$ on the environment e, and $\mathcal{L}(\cdot, \cdot)$: $\mathbb{Y} \times \mathbb{Y} \to \mathbb{R}$ denotes a loss function. We further decompose $f(\cdot) = \rho \circ h$, where the $h(\cdot) : \mathbb{G} \to \mathbb{R}^d$ is a GNN encoder, $d$ is the hidden dimension and $\rho : \mathbb{R}^d \to \mathbb{Y}$ is the classifier. It is known that the Eq. 1 is difficult to solve since environment labels are unobserved. Hence, previous works focus on inferring the environment labels (Wu et al., 2022; Li et al., 2022b; Yang et al., 2022). For example, GIL (Li et al., 2022b) clusters variant subgraphs of all graphs by K-means to infer the environments. However, due to the high heterogeneity of environments in the training dataset, GIL (Li et al., 2022b) will suffer from two limitations. First, without the constraints of class numbers in each environment, GIL (Li et al., 2022b) tends to cluster the same class graphs into an environment, which fails to learn invariant features. Second, when the wrong variant subgraphs are identified, GIL also fails to cluster graphs with the same variant subgraphs into an environment. More details are given in Sec. 5.2 where

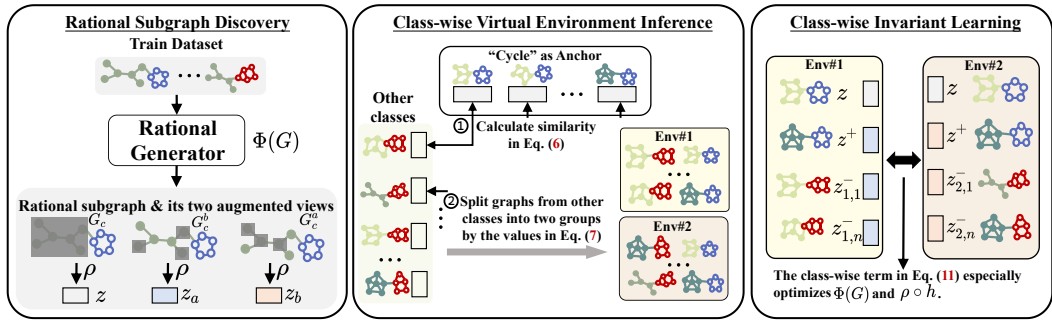

Figure 2: The overall architecture of C-VEI. The core modules are Rationale Subgraph Discovery (RSD), Class-wise Virtual Environment inference (CVE), and Class-wise Invariant Learning (CIL). First, a GNN-based subgraph generator $\Phi$ identifies the invariant subgraph $G_c$ and its augmented views $G_c^a$ and $G_c^b$. Then, the GNN encoder $\rho$ encodes $G_c$, $G_c^a$ and $G_c^b$ into representations. Based on the representations, CVE generates class-wise virtual environments for each class. A class-wise IRM objective is applied to jointly optimize all the components.

the experiments on SP-Motif reveal the limitations of GIl (Li et al., 2022b). Instead of inferring the actual environments and learning environment-wise invariance (Li et al., 2022b; Wu et al., 2022; Chen et al., 2022b), we propose to infer *class-wise virtual environments*, which is easier to learn by leveraging labels. Moreover, we apply a class-wise invariant risk minimization to disregard class-wise spurious correlations and preserve class-wise invariance. Later, we will introduce our C-VEI in detail.

## 4 METHOD

In this section, we introduce our proposed method (see Fig. 2) in detail. Specifically, C-VEI comprises three components. First, C-VEI applies a rationale generator (RG) to discover the invariant subgraph. Then, a class-wise virtual environment construction (CVE) module is designed to heuristically construct class-wise environments. Next, C-VEI optimizes class-wise IRM based on the contrastive objective to generate graph representations that can generalize to test graphs under domain shift.

**Rationale Subgraph Discovery.** In this work, we assume that each input graph $(G, Y) \in \mathcal{D}$ consists of two disjoint parts: the invariant subgraph $G_c \in G$ and the variant subgraph $G_s \in G$. The invariant subgraph $G_c$ has an invariant relationship with the label across different environments while $G_s$ has a spurious correlations with labels. Therefore, identifying invariant subgraphs is helpful for graph OOD generalization. Here, we apply a rationale generator $\Phi(\cdot)$ to predict the invariant subgraph as $G_c = \Phi(G)$. Following previous invariant learning works (Chen et al., 2022b; Li et al., 2022b; Wu et al., 2022), an optimal invariant subgraph generator $\Phi^*(\cdot)$ for graph $G$ should satisfy two property:

- **Sufficiency property**: $Y = h^*(\rho^*(\Phi^*(G))) + \varepsilon, \varepsilon \perp G$, where $\rho^*(\cdot)$ denotes the optimal GNN encoder, $h^*$ is the optimal classifier, $\perp$ indicates statistical independence, and $\varepsilon$ is random noise.
- **Invariance property**: $\forall e^1, e^2 \in \mathcal{E}_{\text{all}}, \mathbb{P}^{e^1}(Y|\Phi^*(G)) = \mathbb{P}^{e^2}(Y|\Phi^*(G))$.

The sufficiency property means that the generated invariant subgraphs should have sufficient predictive abilities in predicting the graph labels. The invariance property means that an optimal subgraph generator satisfies a good generalization, *i.e.,* it can generate accurate invariant subgraphs across different environments.

Specifically, given an input graph instance $G = (\mathcal{V}, \mathcal{E})$ with the node set $\mathcal{V}$ and the edge set $\mathcal{E}$, its adjacency matrix is $\mathbf{A} \in [0,1]^{|\mathcal{V} \times \mathcal{V}|}$, where $\mathbf{A}_{ij} = 1$ denotes the edge from node $i$ to node $j$, and $\mathbf{A}_{ij} = 0$ otherwise. The rationale generator $\Phi$ first adopts a GNN (denoted by $\mathbf{GNN}_1$) to generate the mask matrix $\mathbf{M} \in \mathbb{R}^{|\mathcal{V} \times \mathcal{V}|}$ on $\mathbf{A}$, where mask $M_{i,j}$ indicates the importance of edge $A_{ij}$:

$$\mathbf{Z} = \mathbf{GNN}_1(g), \quad \mathbf{M}_{i,j} = \sigma(\mathbf{Z}_i^T \mathbf{Z}_j), \tag{2}$$

where $\sigma(\cdot)$ is the sigmoid function and $\mathbf{Z} \in \mathbf{R}^{|\mathcal{V} \times d|}$ summarizes the $d$-dimensional representations of all nodes. The generator then selects the edges with the highest masks to construct the rationale $G_c$ and collects $G_c$'s complement as $G_s$, as follows:

$$\mathcal{E}_{G_c} = \text{Top}_\alpha(\mathbf{M} \odot \mathbf{A}), \quad \mathcal{E}_{G_s} = \text{Top}_{1-\alpha}((\mathbf{1} - \mathbf{M}) \odot \mathbf{A}), \tag{3}$$

where $\mathcal{E}_{G_c}$ and $\mathcal{E}_{G_s}$ are the edge sets of $G_c$ and $G_s$, respectively; $\text{Top}_\alpha(\cdot)$ selects the top-$K$ edges with $K = \alpha \times \mathcal{E}$, and $\alpha$ is the hyper-parameter (*e.g.,* 0.6); $\odot$ is the element-wise product. With the selected edges, we can distill the nodes appearing in the edges to establish $G_c$ and $G_s$. Inspired by RCL (Li et al., 2022c), we randomly samples half edges $\mathcal{E}_{G_s^a}$ from complement $\mathcal{E}_{G_s}$ to augment rationale graph and also for the left edges $\mathcal{E}_{G_s^b} = \mathcal{E}_{G_s}/\mathcal{E}_{G_s^a}$. Then we can obtain two augmented view $G_c^a$ and $G_c^b$ as follows:

$$\mathcal{E}_{G_c^a} = \mathcal{E}_{G_c} \cup \mathcal{E}_{G_s^a}, \quad \mathcal{E}_{G_c^b} = \mathcal{E}_{G_c} \cup \mathcal{E}_{G_s^b},$$
$$s.t. \quad \mathcal{E}_{G_s^a}, \mathcal{E}_{G_s^b} \subset \mathcal{E}_{G_s}, \quad \mathcal{E}_{G_s} = \mathcal{E}_{G_s^a} \cup \mathcal{E}_{G_s^b}, \quad |\mathcal{E}_{G_s^a}| = |\mathcal{E}_{G_s^b}| = 0.5 \times |\mathcal{E}_{G_s}|, \tag{4}$$

where $\mathcal{E}_{G_c^a}$ and $\mathcal{E}_{G_c^b}$ are the edge sets of $G_c^a$ and $G_c^b$, respectively. In the following, we jointly optimize invariant subgraph generator $\Phi$, GNN encoder $\rho$, and classifier $h$ to satisfy the mentioned **invariance** and **sufficiency** property.

**Class-wise Virtual Environment Inference.** In this paper, we aim to optimize the objective function in Eq. 1 to obtain invariant graph representation. However, Eq. 1 is difficult to optimize as we do not have environment annotations. To solve this problem, we propose an efficient Class-wise Virtual Environment inference (CVE) module, which aims to heuristically create two virtual environments $\mathcal{E}'_{tr}$ for each class by similarity. Specifically, for an anchor class $c$ containing $k$ graphs, environment Env#1 contains these $k$ samples as positive and the "similar" samples from other classes as negative; environment Env#2 contains the same positive samples while the "dissimilar" samples from other classes as negative. A straightforward way to define the "similarity" between two samples is to use cosine similarity. Thus, we compute the cosine similarity between each rationale subgraph pair sampled from the anchor class and other classes receptively, formulated as:

$$z^c = \rho(G_c^c) \in \mathbb{R}^{k,d}, \qquad \{G_c^c \in G^c | (G^c, Y^c) \in \mathcal{D}, Y^c = c\}$$
$$z^o = \rho(G_c^o) \in \mathbb{R}^{n,d}, \qquad \{G_c^o \in G^c | (G^o, Y^o) \in \mathcal{D}, Y^o \neq c\} \tag{5}$$
$$\mathbf{S} = z^o \cdot z^{cT} \in \mathbb{R}^{n \times k},$$

where $z^c$ and $z^o$ are respectively the normalized feature of invariant subgraph from anchor class and other classes and $n$ is the number of graphs in other classes. Meanwhile, due to the similarities of different classes being different, to remove the effect of class from $\mathbf{S}$, we adjust every sample-to-sample similarity by subtracting a class-to-class similarity. Specifically, we first calculate the class feature by averaging all rationale subgraph features in each class. Next, the class-to-class similarity $\mathcal{M} \in \mathbb{R}^{C \times C}$ is calculated by the cosine similarity among classes, and $C$ is the number of classes in the dataset. We then can obtain a purer similarity which has environment effects:

$$\tilde{\mathbf{S}}_{\mathbf{i,j}} = z^o[i] \cdot z^c[j] - \mathcal{M}[Y^o[i], c]. \tag{6}$$

Then, we average this similarity matrix along the axis of the anchor class, as follows:

$$\mathbf{L} = \frac{1}{k} \sum_{j=1}^{k} \tilde{\mathbf{S}}_{\mathbf{i,j}}, \quad \mathbf{L} \in \mathbb{R}^n. \tag{7}$$

Finally, it is easy to get "similar" samples (corresponding to lower half values in $\mathbf{L}$ grouped in Env#1 and "dissimilar" samples (corresponding to the higher half values in $\mathbf{L}$) grouped in Env#2.

**Invariant Representation Learning.** With the automatically constructed class-wise virtual environments, we are ready to remove the spurious correlations by optimizing a class-wise IRM objective function. For each anchor class $k$, we define an environment-based supervised contrastive loss. Specifically, our loss is computed within each environment $e \in \mathcal{E}_c = e_1, e_2$. For an anchor class graph in $e_1$, we take the representations of other rationale subgraphs $G_c$ in the anchor class and their augmented views $G_c^a$ (or $G_c^b$ for $e = e_2$) as positive $\mathbf{z}^+$ and the representations of rationale subgraphs from other class samples as negative $\mathbf{z}^-$. Then we have:

$$\mathcal{L}(e \in \mathcal{E}_k, w = 1) = \sum_{\mathbf{z} \in e} \frac{1}{N^+} \sum_{\mathbf{z}^+ \in e} [-log \frac{\exp(\mathbf{z}^T \mathbf{z}^+ \cdot w)}{\exp(\mathbf{z}^T \mathbf{z}^+) + \sum_{\mathbf{z}^- \in e} \exp(\mathbf{z}^T \mathbf{z}^-)}], \tag{8}$$

where $N^+$ denotes the number of the positive samples in the current mini-batch and $w = 1$ is a "dummy" classifier to calculate the gradient penalty term (Koyama & Yamaguchi, 2020). Therefore,

the proposed class-wise IRM loss is:

$$\mathcal{L}_{irm}^{class} = \sum_{k=1}^{C} \mathcal{L}_{irm}^{k} = \sum_{k=1}^{C} \sum_{e \in \mathcal{E}_k} \mathcal{L}(e, w=1) + \text{trace}(\text{Var}(\triangledown \mathcal{L}(e, w=1))). \tag{9}$$

Meanwhile, we follow GIL (Li et al., 2022b) and apply a conventional IRM loss objective function to constrain the rationale subgraphs and its augmented view to satisfy **invariance** and **sufficiency**:

$$\mathcal{L}_{irm}^{ce} = \sum_{G_{sub} \in \{G_c, G_c^a, G_c^b\}} \mathcal{L}_{ce}(G_{sub}) + \text{trace}(\text{Var} \triangledown \mathcal{L}_{ce}(G_{sub})). \tag{10}$$

Finally, the overall training objective is the combination of the conventional cross entropy $\mathcal{L}_{ce}$ and the class-wise IRM regularization $\mathcal{L}_{irm}^{class}$:

$$\min_{\rho, h, \Phi} \mathcal{L}_{irm}^{ce} + \lambda \mathcal{L}_{irm}^{class}, \tag{11}$$

where $\lambda$ is the trade-off hyper-parameter. The former term in Eq. 11 forces the rationale subgraphs to satisfy **invariance** and **sufficiency** property mentioned in Sec. 4. Meanwhile the latter term in Eq. 11 aims to remove spurious correlation on each class by contrasting similar but different class representations in latent space. In our experiments, we demonstrate that the two components are mutually beneficial for an effective graph OOD generalization framework. Moreover, we directly use $\rho \circ h(\Phi(G))$ for inference.

## 5 EXPERIMENTS

### 5.1 EXPERIMENT SETUP

**Datasets.** We adopt one synthetic dataset with controllable ground-truth environments and four real-world benchmarks for the graph classification task.

- **SP-Motif dataset.** Following Chen et al. (2022b); Li et al. (2022b); Wu et al. (2022), we construct 3-class synthetic datasets, where each graph consists of one variant subgraph and one rationale subgraph, *i.e.,*, motif. The variant subgraph includes *Tree*, *Ladder*, and *Wheel* (denoted by $V = 0$, 1, 2, respectively), and the invariant subgraph includes *Cycle*, *House*, and *Crane* (denoted by $I = 0, 1, 2$). The ground-truth label $Y$ only depends on the invariant subgraph $I$, which is sampled uniformly. The spurious correlation between $V$ and $Y$ is injected by controlling the variant subgraphs distribution as $P(V) = r$ if $V = I$ and $P(V) = (1 - r)/2$ if $V \neq I$. Intuitively, $r$ controls the strength of the spurious correlation. We set $r$ to different values in the testing and training set to simulate the distribution shifts
- **Graph-SST5 dataset.** Following Chen et al. (2022b), we split the data curated from sentiment graph data to study distribution shifts in graph sizes. We convert sentiment sentence classification datasets **SST5** into graphs as Graph-SST5 dataset.
- **SST-Twitter dataset.** Similar to the Graph-SST5 dataset, we convert sentiment sentence classification datasets **SST-Twitter** (Socher et al., 2013; Dong et al., 2014) into graphs to study the distribution shifts in graph sizes.
- **DrugOOD datasets.** To evaluate the OOD performance in realistic scenarios, we also include three datasets from the DrugOOD benchmark (Ji et al., 2023). In particular, we select `DrugOOD-lbap-core-ic50-assay/scaffold/size` from Ligand Based Affinity Prediction task which uses `ic50` measurement type and contains `core` level annotation noises.

**Evaluation.** We report the classification accuracy for all datasets, except for DrugOOD datasets where we use ROC-AUC following Li et al. (2022b); Chen et al. (2022b). We repeat the evaluation multiple times, select models based on the validation performances, and report the mean and standard deviation of the corresponding metric.

**Baseline.** We thoroughly compare C-VEI with Empirical Risk Minimization (ERM) (Vapnik, 1991) and the following two categories of baselines:

- **Interpretable Baselines.** GIB (Yu et al., 2021), ASAP Pooling (Ranjan et al., 2020), GAST (Miao et al., 2022) and DIR (Wu et al., 2022). We compare with SOTA interpretable GNNs to validate the effectiveness of the optimization objective in C-VEI.

Table 1: OOD generalization performance on complex distribution shifts for real-world graphs. **Bold** numbers are superior results.

| | Scenario 1: $r_{test} = 1/3$ | | | | | Scenario 2: $r_{test} = 0.2$ | | | | |
|---|---|---|---|---|---|---|---|---|---|---|
| $r_{train}$ | 0.33 | 0.5 | 0.6 | 0.7 | 0.8 | 0.33 | 0.5 | 0.6 | 0.7 | 0.8 |
| IRM | $52.00_{\pm2.34}$ | $50.60_{\pm3.54}$ | $47.84_{\pm6.95}$ | $38.80_{\pm3.72}$ | $39.84_{\pm3.21}$ | $50.24_{\pm6.73}$ | $41.60_{\pm4.75}$ | $35.24_{\pm5.35}$ | $34.92_{\pm8.03}$ | $29.44_{\pm5.47}$ |
| v-REx | $53.16_{\pm3.25}$ | $46.04_{\pm6.11}$ | $45.36_{\pm3.66}$ | $40.24_{\pm3.86}$ | $39.48_{\pm3.00}$ | $50.56_{\pm2.83}$ | $37.16_{\pm6.24}$ | $34.52_{\pm3.00}$ | $29.72_{\pm4.58}$ | $27.32_{\pm3.18}$ |
| GroupDRO | $53.20_{\pm4.91}$ | $51.40_{\pm4.35}$ | $48.32_{\pm5.35}$ | $39.12_{\pm4.27}$ | $38.40_{\pm2.76}$ | $52.68_{\pm4.04}$ | $43.68_{\pm4.05}$ | $31.92_{\pm6.84}$ | $34.36_{\pm8.41}$ | $28.88_{\pm5.14}$ |
| ERM | $53.60_{\pm3.79}$ | $51.24_{\pm4.13}$ | $47.04_{\pm7.01}$ | $38.80_{\pm3.72}$ | $37.84_{\pm3.01}$ | $48.48_{\pm4.53}$ | $41.72_{\pm4.81}$ | $36.92_{\pm6.93}$ | $35.72_{\pm8.33}$ | $28.80_{\pm3.91}$ |
| DIR | $52.96_{\pm5.06}$ | $52.08_{\pm1.93}$ | $50.12_{\pm2.76}$ | $49.84_{\pm2.46}$ | $45.20_{\pm1.11}$ | $50.68_{\pm5.20}$ | $49.96_{\pm1.75}$ | $45.44_{\pm6.00}$ | $40.56_{\pm2.36}$ | $39.92_{\pm4.53}$ |
| GSAT | $53.67_{\pm3.65}$ | $53.34_{\pm4.08}$ | $51.54_{\pm3.78}$ | $50.12_{\pm3.29}$ | $45.83_{\pm4.01}$ | $51.36_{\pm4.21}$ | $50.48_{\pm3.98}$ | $46.93_{\pm5.03}$ | $43.55_{\pm3.67}$ | $40.35_{\pm4.21}$ |
| GIL | $55.44_{\pm3.11}$ | $54.56_{\pm3.02}$ | $53.60_{\pm4.82}$ | $53.12_{\pm2.18}$ | $51.24_{\pm3.88}$ | $54.80_{\pm3.93}$ | $52.48_{\pm4.41}$ | $50.08_{\pm5.47}$ | $47.44_{\pm2.87}$ | $46.36_{\pm3.80}$ |
| Ours | $\mathbf{73.40_{\pm1.01}}$ | $\mathbf{68.28_{\pm0.90}}$ | $\mathbf{69.28_{\pm3.32}}$ | $\mathbf{63.78_{\pm5.54}}$ | $\mathbf{62.22_{\pm10.93}}$ | $\mathbf{72.90_{\pm1.02}}$ | $\mathbf{69.26_{\pm0.69}}$ | $\mathbf{66.52_{\pm1.91}}$ | $\mathbf{62.30_{\pm4.65}}$ | $\mathbf{64.74_{\pm7.89}}$ |

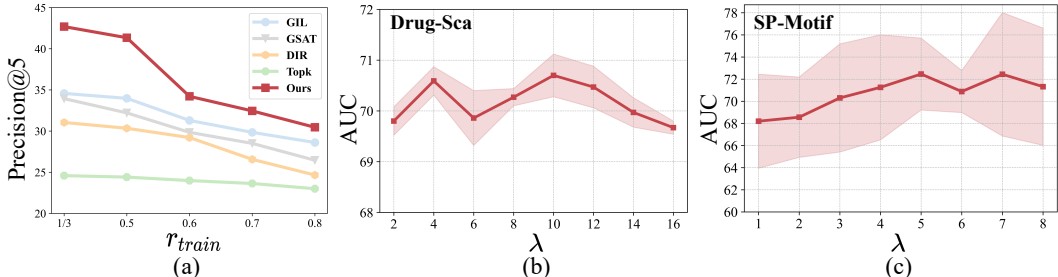

Figure 3: (a) The results of discovering the ground truth invariant subgraphs on SP-Motif ($r_{test} = 0.2$); (b) The hyper-parameter sensitivity analysis on Drug-Scaffold; (c) The hyper-parameter sensitivity analysis on SP-Motif.

- **Invariant Learning Baselines.** GroupDRO (Sagawa et al., 2019), IRM (Arjovsky et al., 2019), v-REx (Krueger et al., 2021), IB-IRM (Ahuja et al., 2021), CNC (Zhang et al., 2022), CIGA (Chen et al., 2022b), LSI (Yang et al., 2022), GIL (Li et al., 2022b), and GALA (Chen et al., 2023). This class of algorithms improves the robustness and generalization of GNNs, which helps the models better generalize in unseen groups or out-of-distribution datasets.

## 5.2 EXPERIMENT ON SP-MOTIF

**Setting.** We first compare our C-VEI with state-of-the-art methods on the synthetic SP-Motif dataset. Following Li et al. (2022b), we introduce variations in $r$ within both the training and testing datasets of SP-Motif to simulate different levels of distribution shifts. For the training set, we select $r_{train}$ from 1/3, 0.5, 0.6, 0.7, 0.8, 0.9. A higher value of $r_{train}$ indicates a stronger spurious correlation between Y and $G_V$ in the training set, while $r_{train} = 1/3$ implies a balanced training set without any spurious correlation. For the testing set, we consider two settings: (1) $r_{test} = 1/3$, which simulates random attachment of invariant and variant subgraphs without spurious correlations; (2) $r_{test} = 0.2$, indicating the presence of reversed spurious correlations in the testing set, posing a greater challenge.

**C-VEI has better generalization ability than baselines.** The results are reported in Tab. 1. It is clear that C-VEI consistently outperforms all state-of-the-art methods across all datasets. Specifically, C-VEI surpasses the current best competitor GIL (Li et al., 2022b) by 17.9% with $r_{train} = 0.33$ and $r_{test} = 0.2$. The results demonstrate that our proposed C-VEI has a remarkable graph OOD generalization on unknown domain shifts. As the degree of the distribution shift is large, invariant baselines show more stable performance. Among them, GIL is a graph invariant learning method for OOD generalization, which is one competitive baseline. As an interpretable graph method, GSAT also achieves promising results. Meanwhile, our proposed C-VEI achieves optimal performance. It indicates that learning invariant subgraphs is critical and beneficial for OOD generalization on graphs.

**C-VEI has better intrinsic interpretability than baselines.** Following Li et al. (2022b), to analyze whether our proposed C-VEI can accurately capture the rationale subgraph, we compare C-VEI with baselines that also output subgraphs using the ground-truth invariant subgraph on SP-Motif ($r_{test} = 0.2$). The Precision@5 is reported in Fig. 3(a). From the consistent improvements over the baselines, we find that C-VEI has better intrinsic interpretability than the baselines. As we can see, the method with better interpretation tends to achieve better OOD generalization performance. It also

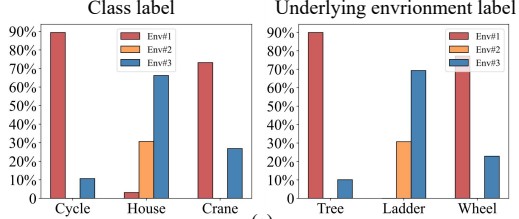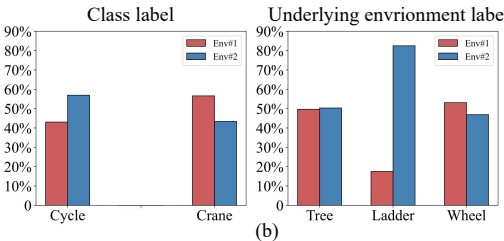

(a)     (b)

Figure 4: The distributions of the class label and underlying environment label on environment learned by (a) GIL (Li et al., 2022b) and (b) C-VEI. For C-VEI, considering the positive samples are same in two environments, we only investigate negative samples.

Table 2: OOD generalization performance on complex distribution shifts for real-world graphs. Numbers in bold represent the best results.

| Datasets Methods | Drug-Assay | Drug-Sca | Drug-Size | Graph-SST5 | SST-Twitter |
|---|---|---|---|---|---|
| IRM (Arxiv2019) | $72.12_{\pm 0.49}$ | $68.69_{\pm 0.65}$ | $66.54_{\pm 0.42}$ | $43.69_{\pm 1.26}$ | $63.50_{\pm 1.23}$ |
| v-REx (ICML2021) | $72.05_{\pm 1.25}$ | $68.92_{\pm 0.98}$ | $66.33_{\pm 0.74}$ | $43.28_{\pm 0.52}$ | $63.21_{\pm 1.57}$ |
| EIIL (ICML2021) | $72.60_{\pm 0.47}$ | $68.45_{\pm 0.53}$ | $66.38_{\pm 0.66}$ | $42.98_{\pm 1.03}$ | $62.76_{\pm 1.72}$ |
| IB-IRM (NeurIPS2021) | $72.50_{\pm 0.49}$ | $68.50_{\pm 0.40}$ | $66.64_{\pm 0.28}$ | $40.85_{\pm 2.08}$ | $61.26_{\pm 1.20}$ |
| CNC (ICML2022) | $72.40_{\pm 0.46}$ | $67.24_{\pm 0.90}$ | $65.79_{\pm 0.80}$ | $42.78_{\pm 1.53}$ | $61.03_{\pm 2.49}$ |
| ERM (NeurIPS1991) | $71.79_{\pm 0.27}$ | $68.85_{\pm 0.62}$ | $66.70_{\pm 1.08}$ | $43.89_{\pm 1.73}$ | $60.81_{\pm 2.05}$ |
| GIB (ICLR2021) | $63.01_{\pm 1.16}$ | $62.01_{\pm 1.41}$ | $55.50_{\pm 1.42}$ | $38.64_{\pm 4.52}$ | $48.08_{\pm 2.27}$ |
| DIR (ICLR2022) | $68.25_{\pm 1.40}$ | $63.91_{\pm 1.36}$ | $60.40_{\pm 1.42}$ | $41.12_{\pm 1.96}$ | $59.85_{\pm 2.98}$ |
| CIGA (NeurIPS2022) | $73.17_{\pm 0.39}$ | $69.70_{\pm 0.27}$ | $67.78_{\pm 0.76}$ | $44.91_{\pm 4.31}$ | $64.45_{\pm 1.99}$ |
| GALA (ICLRw2023) | - | - | - | $44.8_{\pm 1.02}$ | $62.45_{\pm 0.62}$ |
| LSI (NeurIPS2022) | $71.38_{\pm 0.68}$ | $68.02_{\pm 0.55}$ | $66.51_{\pm 0.55}$ | - | - |
| Ours | $\mathbf{73.60_{\pm 0.35}}$ | $\mathbf{70.59_{\pm 0.21}}$ | $\mathbf{67.80_{\pm 0.33}}$ | $\mathbf{45.24_{\pm 1.17}}$ | $\mathbf{64.78_{\pm 0.84}}$ |

verifies our assumption that accurately capturing rationale subgraphs is beneficial for graph OOD generalization.

**Analysis of learned class-wise virtual environment.** We further investigate the class-wise environments learned by C-VEI and the actual environments inferred by GIL (Li et al., 2022b) on the SP-Motif dataset ($r_{train} = 0.8$, $r_{test} = 0.33$). Specifically, we investigate the distribution of labels and underlying environment labels in each environment[1]. For simplicity, for C-VEI, we only show two class-wise environments learned by C-VEI for the "*House*" class. The results are shown in Fig. 4. For GIL, since it aims to infer the actual environments, each inferred environment should ideally be split according to environment labels, *i.e.*, each environment contains graphs from multiple classes but with the same base subgraph. However, we find that the environments inferred by GIL are unreliable and suffer from two limitations: i) without extra constraints, some environments only contain graphs from a single class, which fails to remove spurious correlations (see left of Fig. 4(a), Env#2 only contains "*House*" class samples); ii) the inferred environments are sensitive to variant graph identification. When variant subgraphs are misidentified, GIL fails to cluster graphs with the same variant subgraphs (see Env#1 and Env#3 in right of Fig. 4(a)). While, instead of directly inferring actual environments, we propose to generate class-wise virtual environments which should be reliable and effective to remove the spurious correlation in each class. As the right of Fig. 4(b) shows, the graphs with environment label "*Tree*" and "*Wheel*" are evenly distributed in Env#1 and Env#2 but graphs with "*Ladder*" as environment label are mostly split in Env#2. Considering the difference of "*Ladder*" in two class-wise environments, it is easy to decouple the spurious correlation between the "*House*" class and the "*Ladder*" base.

## 5.3 EXPERIMENTS ON REAL-WORLD GRAPH DATASETS

We further evaluate the effectiveness of our method on real-world graph datasets. The results are reported in Tab. 2. Our C-VEI achieves the best performance on all datasets, indicating that C-VEI can handle distribution shifts on real-world graphs. For example, C-VEI increase the classification

---

[1]Here, we set the number of environments for C-VEI and K-means as $C$ and $2C$ respectively, where the class number $C = 3$.

Table 3: The ablation study of each component on DrugOOD datasets. Numbers in bold represent the best results.

| Methods | Drug-Assay | Drug-Scaffold | Drug-Size |
|---|---|---|---|
| Ours w/o rationale generator | $72.46_{\pm 0.12}$ | $69.28_{\pm 0.40}$ | $67.05_{\pm 0.19}$ |
| Ours w/o $\mathcal{L}_{irm}^{class}$ | $72.83_{\pm 0.21}$ | $70.00_{\pm 0.45}$ | $67.45_{\pm 0.28}$ |
| Ours w/ random environment | $70.36_{\pm 0.21}$ | $68.60_{\pm 0.19}$ | $66.67_{\pm 0.29}$ |
| Ours w/ inferred actual environment | $71.32_{\pm 0.43}$ | $68.73_{\pm 0.32}$ | $66.92_{\pm 0.18}$ |
| Ours | $\mathbf{73.60_{\pm 0.35}}$ | $\mathbf{70.59_{\pm 0.21}}$ | $\mathbf{67.80_{\pm 0.33}}$ |

accuracy by 0.89% on Drug-Scaffold against the strongest baselines respectively. Besides, different datasets have different distribution shifts, *e.g.,* Graph-SST5 has different node degrees, and the distribution shift of DrugOOD (Ji et al., 2023) is size, assay, and scaffold. Therefore, the results show that our proposed C-VEI is robust against diverse types of distribution shifts in real graphs.

## 5.4 ABLATION STUDY

We analyze the contribution of each component to the final performance in this section. Tab. 3 reports detailed ablation experimental results on Drug-Assay, Drug-Scaffold, and Drug-Size.

**Effect of the rationale generator.** To study the impact of the rationale generator, we remove the rationale generator from C-VEI and use the representation of the full graph to infer the virtual environment and classify. As shown in Tab. 3, removing the rationale generator leads to performance degradation. This verifies our assumption that the rationale graph identification is important to boosting performance. With the aid of such a rationale generator, the impact of our proposed class-wise virtual environment inference and class-wise IRM regularization can be further strengthened.

**Effect of the class-wise virtual experiment inference.** To evaluate the impact of our proposed class-wise virtual experiment inference module, we compare our C-VEI with two variants: i) we randomly split graphs into $2C$ environments; ii) we follow GIL (Li et al., 2022b) to infer actual environment by carrying out K-means on the representations of variant subgraphs, where the number of environments is $2C$ and $C$ is the number of class. Our C-VEI beats variant i) and ii) across all datasets (see Tab. 3). It reveals that a meaningful and effective environment is essential for graph invariant learning. Thus, we can attribute the main superiority of our full model to the inferred class-wise virtual experiment.

**Effect of the class-wise IRM regularization.** Similarly, we remove the proposed class-wise IRM regularization to evaluate its impact. As shown in Tab. 3, only removing the class-wise IRM regularization degrades the OOD generalization performance. This indicates that the new learning objective can better guide the model to learn invariant representation against distribution shift.

## 5.5 HYPER-PARAMETER SENSITIVITY ANALYSIS

In this section, we investigate the sensitivity of our C-VEI to the only hyper-parameters: the trading-off parameter $\lambda$ in Eq. 11. We carry out experiments on Drug-Scaffold and SP-Motif datasets, and the results are reported in Fig. 3(b) and Fig. 3(c) respectively. As we can see, our C-VEI are robust to the different values of $\lambda$ across different datasets and distribution shifts.

## 6 CONCLUSION

In this paper, we identify several practical challenges in the underlying environment inference (Li et al., 2022b; Yang et al., 2022; Chen et al., 2022b; Wu et al., 2022) for graph invariant learning. To tackle the challenge, instead of inferring underlying actual environments, we introduce a novel graph invariant learning framework, named by C-VEI, which infers *class-wise virtual environments* with the assistance of labels and learning class-wise invariant features. We conduct extensive experiments on several graph OOD generalization benchmarks and show that our C-VEI achieves new state-of-the-art results. Identifying class-wise spurious characteristics across classes under the heterogeneous environment, positions our approach as a noteworthy contribution to graph representation learning.

## 7 ETHICS STATEMENT

In this work, we propose a novel algorithm for graph invariant learning, where no human subject is related. We believe the graphs OOD generalization is beneficial for inspecting and eliminating potential discrimination and fairness issues in deep models for real applications.

## 8 REPRODUCIBILITY STATEMENT

We summarize the efforts made to ensure reproducibility in this work. (1) Datasets: we use one synthetic dataset and three public datasets where the processing details are included in Appendix A.1. (2) Model Training: We provide the training details (including hyper-parameters settings) in Appendix A.2 and the procedure of training in Algorithms A.3.

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

# Appendix of C-VEI

Table 4: Information about the datasets used in experiments. The number of nodes and edges are taking average among all graphs.

| DATASETS | # TRAINING | # VALIDATION | # TESTING | # CLASSES | # NODES | # EDGES | METRICS |
|---|---|---|---|---|---|---|---|
| SP-MOTIF | 9,000 | 3,000 | 3,000 | 3 | 44.96 | 65.67 | ACC |
| SST5 | 6,090 | 1,186 | 2,240 | 5 | 19.85 | 37.70 | ACC |
| TWITTER | 3,238 | 694 | 1,509 | 3 | 21.10 | 40.20 | ACC |
| DRUGOOD-ASSAY | 34,179 | 19,028 | 19,032 | 2 | 32.27 | 70.25 | ROC-AUC |
| DRUGOOD-SCAFFOLD | 21,519 | 19,041 | 19,048 | 2 | 29.95 | 64.86 | ROC-AUC |
| DRUGOOD-SIZE | 36,597 | 17,660 | 16,415 | 2 | 30.73 | 66.90 | ROC-AUC |

## A   IMPLEMENTATION DETAILS

### A.1   DETAILS ABOUT THE DATASETS

We provide more details about the datasets (see Tab. 4) that are used in our experiments.

**SP-Motif datasets.** Following Chen et al. (2022b); Li et al. (2022b), we construct 3-class synthetic datasets based on BAMotif (Ying et al., 2019; Luo et al., 2020), where the model needs to tell which one of three motifs (House, Cycle, Crane) that the graph contains. For each dataset, we generate 3000 graphs for each class at the training set, 1000 graphs for each class at the validation set and testing set, respectively. During the construction, we respectively inject the different distribution shifts in the training data and the testing/validation data with $r_{train}$ and $r_{test}$. Following Chen et al. (2022b); Li et al. (2022b), the motif and one of the three base graphs (Tree, Ladder, Wheel) are artificially (spuriously) correlated with a probability of various biases, and equally correlated with the other two. Specifically, given a predefined bias $r$ ($r \in \{r_{train}, r_{test}\}$), the probability of a specific motif (e.g., House) and a specific base graph (Tree) will co-occur is $r$ while for the others is $(1 - r)/2$ (e.g., House-Ladder, House-Wheel).

**Graph-SST datasets.** Following Chen et al. (2022b), we split the data curated from sentiment graph data (Yuan et al., 2020), that converts sentiment sentence classification datasets **SST5** and **SST-Twitter** (Socher et al., 2013; Dong et al., 2014) into graphs, where node features are generated using BERT (Devlin et al., 2019) and the edges are parsed by a Biaffine parser (Gardner et al., 2018). Our splits are created according to the averaged degrees of each graph. Specifically, we assign the graphs as follows: Those that have smaller or equal than 50-th percentile averaged degree are assigned into training, those that have averaged degree large than 50-th percentile while smaller than 80-th percentile are assigned to validation set, and the left are assigned to test set. For SST5 we follow the above process while for Twitter we conduct the above split in an inversed order to study the OOD generalization ability of GNNs trained on large degree graphs to small degree graphs.

**DrugOOD datasets.** To evaluate the OOD performance in realistic scenarios with realistic distribution shifts, we also include three datasets from DrugOOD benchmark. DrugOOD is a systematic OOD benchmark for AI-aided drug discovery, focusing on the task of drug target binding affinity prediction for both macromolecule (protein target) and small-molecule (drug compound). The molecule data and the notations are curated from realistic ChEMBL database (Mendez et al., 2019). Complicated distribution shifts can happen on different assays, scaffolds and molecule sizes. In particular, we select `DrugOOD-lbap-core-ic50-assay`, `DrugOOD-lbap-core-ic50-scaffold`, and `DrugOOD-lbap-core-ic50-size`, from the task of Ligand Based Affinity Prediction which uses `ic50` measurement type and contains `core` level annotation noises. For more details, we refer interested readers to Ji et al. (2023).

## A.2 TRAINING AND OPTIMIZATION IN EXPERIMENTS

During the experiments, we do not tune the hyperparameters exhaustively while following the common recipes for optimizing GNNs. Details are as follows.

**GNN encoder.** For fair comparison, we use the same GNN architecture as graph encoders for all methods. Following Chen et al. (2022b), we use 3-layer GNN with Batch Normalization (Ioffe & Szegedy, 2015) between layers and JK residual connections at last layer (Xu et al., 2018). For the architectures, following Li et al. (2022b); Wu et al. (2022), we use the LeGCN Ranjan et al. (2020) with mean readout for SP-Motif datasets. Meanwhile, we use the GCN (Kipf & Welling, 2017) with mean readout for Graph-Twitter/STT5 where we follow Chen et al. (2022b) and use a GIN (Xu et al., 2019) with max readout for DrugOOD datasets where we follow the backbone used in the paper (Ji et al., 2023), i.e., 4-layer GIN with sum readout. The hidden dimensions are fixed as 32 for SP-Motif and 128 for SST5, Twitter and DrugOOD datasets.

**Optimization and model selection.** Following Chen et al. (2022b), we use Adam optimizer (Kingma & Ba, 2015) with a learning rate of 1e-3 and a batch size of 32 for all models at all datasets. Except for DrugOOD datasets, we use a batch size of 128 following the original paper (Ji et al., 2023). To avoid overfitting, we also employ an early stopping of 5 epochs according to the validation performance.

Meanwhile, dropout (Srivastava et al., 2014) is also adopted for some datasets. Specifically, we use a dropout rate of 0.5 for SST5, Twitter, DrugOOD-Assay and DurgOOD-Scaffold and 0.1 for DrugOOD-Size following the practice of Chen et al. (2022b).

**The results of baselines.** In this paper, we use the results of baselines reported in Chen et al. (2022b); Li et al. (2022b) for comparisons. Meanwhile, for the analysis of environment inferred by GIL (Li et al., 2022b), we implements it according the pseudo-code in the original paper.

## A.3 IMPLEMENTATIONS OF C-VEI.

For fair comparison, C-VEI uses the same GNN architecture for GNN encoders as the baseline methods. We did not do exhaustive hyperparameters tuning for the loss Eq. 11. By default, we merely search $\lambda$ from $\{0.5, 1, 2, 4, 16, 32\}$ for Drug-OOD and $\{1, 2, 3, 4, 5, 6, 7, 8\}$ for other datasets. Moreover, for a fair comparison, we uniformly set the hyper-parameter $\alpha$ for C-VEI and baselines as 0.25, 0.5, 0.6, and 0.8 on SP-Motif, Graph-SST5, Graph-Twitter and DrugOOD respectively. Meanwhile, The detailed algorithm for C-VEI is given in the Algorithm 1.

---

**Algorithm 1** Pseudo code for the CIGA framework.

---

**Input:** Training graphs and labels $\mathcal{D} = \{G_i, Y_i\}_{i=1}^{N}$; learning rate $\gamma$; loss weights $\lambda$ required by Eq. 11; the number of training epochs $e$; the number of classes $C$; the batch size $B$;

Randomly initialize parameters of $\Phi, h, \rho$;

**for** $i = 1$ **to** $e$ **do**

    Sample a batch of graphs $\{G^j, Y^j\}_{j=1}^{B}$;

    Estimate the invariant subgraph and its two augmented views for the batch: $\{\widehat{G}_c^j, \widehat{G}_c^{a,j}, \widehat{G}_c^{b,j}\}_{j=1}^{B} = \Phi(\{G^j, Y^j\}_{j=1}^{B})$;

    Make predictions based the estimated invariant subgraph and its two augmented views: $\{\widehat{Y}^j\}_{j=1}^{B} = h(\{\widehat{G}_c^j\}_{j=1}^{B}), \{\widehat{Y_a}^j\}_{j=1}^{B} = h(\{\widehat{G}_c^{a,j}\}_{j=1}^{B}), \{\widehat{Y_b}^j\}_{j=1}^{B} = h(\{\widehat{G}_c^{b,j}\}_{j=1}^{B})$ ;

    Calculate the empirical loss $\mathcal{L}_{irm}^{ce}$ in Eq. 10 with $\{\widehat{Y}^j\}_{j=1}^{B}, \{\widehat{Y_a}^j\}_{j=1}^{B}, \{\widehat{Y_b}^j\}_{j=1}^{B}$;

    Fetch the graph representations of invariant subgraphs from $\rho$ as $\{h_{\widehat{G}_c^j}\}_{j=1}^{B}, \{h_{\widehat{G}_c^{a,j}}\}_{j=1}^{B}, \{h_{\widehat{G}_c^{b,j}}\}_{j=1}^{B}$;

    **for** $c = 1$ **to** $C$ **do**

        Select $c$ as the anchor class and split other classes samples into two group according to the Eq. 7; then obtain the inferred class-wise virtual environment.

        Calculate the contrastive loss $\mathcal{L}_{irm}^{class}$ with Eq. 9 in inferred virtual environments, where positive samples and negative samples are respectively anchor class and other classes graphs.

    **end for**

    Calculate the total class-wise IRM loss in the Eq. 8.

    Calculate the total loss in Eq. 11 with the trade-off hyper-parameter $\lambda$;

    Update parameters of $\Phi, h, \rho$ with respect to $\min_{\rho, h, \Phi} \mathcal{L}_{irm}^{ce} + \lambda \mathcal{L}_{irm}^{class}$ as Eq. 11;

**end for**

---

