# OpenReview forum: "Learning Invariant Graph Representations via Virtual Environment Inference"
_ICLR.cc/2024/Conference — ICLR 2024 Conference Withdrawn Submission_

### Official Review · Reviewer_ELup · 2023-10-30

**Soundness:** 2 fair
**Presentation:** 2 fair
**Contribution:** 2 fair
**Rating:** 5
**Confidence:** 4

**Summary:**

This paper investigates out-of-generalization (OOD) problems in graphs, which learns invariant graph representations under different environments. Instead of inferring the underlying environment labels, this paper proposes to generate class-wise virtual environments. Specifically, a contrastive learning strategy is utilized to pull samples from same class but dissimilar graph representations together and push samples from different classes but similar graph representation away. Experimental results on 6 public datasets and 1 synthetic dataset demonstrate that the proposed model can achieve superior performance on various OOD benchmarks.

**Strengths:**

1)	This paper studies the OOD setting on graph related tasks, which is practical in real-world scenarios.
2)	Ablation studies are given to show the effectiveness of the proposed components.
3)	The proposed model seems to perform very well on the synthetic dataset.

**Weaknesses:**

1)	This paper’s technical contribution is only incremental. The used techniques are all from existing works. For examples, the way of constructing $G_c$ and $G_s$ is from GIL; and the class-wise IRM objective can obtained from various existing works like RGCL.
2)	The improvements on public datasets are very limited. As we can see in Table 2, most of the improvements are within 0.5%. With such limited improvements, it is hard to say the model is effective. It is not clear why GSAT and GIL’ performance is missing on the public datasets in Table 2.
3)	More ablation studies should be conducted to show the effectiveness of the proposed components. In Eq (6), it is unclear how class-to-class matrix $M$ will contribute to the model’s performance. In Eq (7), it is not clear what is the threshold for picking similar and dissimilar samples.
4)	In hyper-parameter analysis section, $\lambda$ is not the only hyper-parameters. There are lots of other hyper-parameters in the model. For instance, masking ratio $\alpha$ and augmentation ratio in Eq (4) should also be analyzed.

There are several typos in the paper. I list some of them as follows:

“one rationale subgraph, i.e.,, motif” ->” one rationale subgraph, i.e., motif”

“is important to boosting performance” -> “is important to boost performance”

**Questions:**

1)	It is not clear why GSAT and GIL’ performance is missing on the public datasets in Table 2.
2)	More ablation studies should be conducted to show the effectiveness of the proposed components.
3)	There are lots of other hyper-parameters in the model. For instance, masking ratio $\alpha$ and augmentation ratio in Eq (4) should also be analyzed.

**Details Of Ethics Concerns:**

NA.

---

### Official Review · Reviewer_6tAG · 2023-10-31

**Soundness:** 2 fair
**Presentation:** 2 fair
**Contribution:** 2 fair
**Rating:** 3
**Confidence:** 5

**Summary:**

This paper proposes a novel framework called Class-wise Invariant Risk Minimization via Virtual Environment Inference (C-VEI) for learning invariant graph representations across different environments. The authors address the challenge of inferring underlying environments by generating reliable virtual environments from a class-wise perspective. They introduce a contrastive strategy to infer class-wise virtual environments and design an invariant risk minimization approach to preserve class-wise invariance. Extensive experiments demonstrate the superiority of C-VEI in graph out-of-distribution generalization tasks.

**Strengths:**

1. This paper propose a novel graph invariant learning framework called C-VEI, which infers class-wise virtual environments with the assistance of labels and learning class-wise invariant features.
2. C-VEI addresses practical challenges in underlying environment inference for graph invariant learning and provides insights into the importance of rationale graph identification and class-wise virtual experiment inference.
3. This paper conducted sufficient experiments to verify the claim and provided necessary appendices to further demonstrate and explain details in experiments, implementations and training pipelines.

**Weaknesses:**

1. Very limited contribution. C-VEI is an implemental work based on GIL [1] and RGCL [2]. Specifically, C-VEI and GIL [1] have highly similar problem background, target task, and framework approach. In particular, Figure 1, Chapter 3, sufficient & invariant assumptions, etc., are highly similar to GIL [1], while the “anchor and augmented views” method is much similar to RGCL [2].
2. Unconvincing experiment results. Table 1 illustrates the results of OOD generalization performance on complex distribution shifts for synthetic graph datasets (here is a typo in caption). However, the results of all baselines in Table 1 are directly copied from GIL [1]. And, as the synthetic graph dataset (SP-Motif dataset) used in GIL [1] is not public released and is impossible to reproduce exactly the same as GIL [1]’s, so, it’s definitely unfair to compare C-VEI and the copied results of other baselines from GIL [1]. Accordingly, the conclusion that C-VEI outperforms all baselines is false.
3. Insufficient motivation. C-VEI claims GIL [1] will fail without sufficient data, thus the subgraph partitioning can easily be misled by spurious subgraphs, which further exacerbates the biases in the inferred environments (Section 1). However, the authors only give conceptual examples in Figure 1(b) instead of empirical analysis, which makes the motivation of C-VEI poor and weak.

[1] Learning Invariant Graph Representations for Out-of-Distribution Generalization, NeurIPS 2022.
[2] Let Invariant Rationale Discovery Inspire Graph Contrastive Learning, ICML 2022.

**Questions:**

1. What’s the relation of C-VEI and CIGA [3]? The title for Algorithm 1 is “Pseudo code for the CIGA framework”, which is confusing.
2. Compared to GIL [1] and RGCL [2], what is the advantage of C-VEI for learning invariant graph representations across different environments? It would be great if the authors could provide some examples for explanation.
3. Further explain the meaning of “environment heterogeneity”. Does it mean the same as heterogenous / homogeneous graphs?
4. Table 2 lacks indispensable comparison with the most important baselines GIL [1] and RGCL [2], and better evaluate C-VEL’s performance on more datasets, like MNIST-75sp [4], Graph-SST2 [5], and Open Graph Benchmark (OGB) [6]. In addition, the graph OOD benchmark GOOD [7] is universally applied to evaluate OOD generalization ability. It would be better to evaluate C-VEL in a more overall perspective.
5. Computational complexity and sufficient analysis of C-VEL are not discussed in the main contents.
6. The symbols and annotations used is complicated. Provide further explanations in a summary table.

[3] Learning Causally Invariant Representations for Out-of-Distribution Generalization on Graphs, NeurIPS 2022.
[4] Understanding Attention and Generalization in Graph Neural Networks, arXiv 2019.
[5] Explainability in Graph Neural Networks: A Taxonomic Survey, TPAMI 2020.
[6] Open Graph Benchmark: Datasets for Machine Learning on Graphs, arXiv 2020.
[7] GOOD: A graph out-of-distribution benchmark, NeurIPS 2022.

---

### Official Review · Reviewer_rRhy · 2023-11-01

**Soundness:** 2 fair
**Presentation:** 2 fair
**Contribution:** 3 good
**Rating:** 3
**Confidence:** 4

**Summary:**

The paper studies OOD graph generalization. It generates virtual environments for graph invariant learning. The proposed method C-VEI aims to discard class-wise spurious correlations and preserve class-wise invariance using contrastive learning on the latent space, which pulls samples from the same class but dissimilar graph representations together and pushes samples from different classes but similar graph representations away. Experiments are conducted to evaluate the method.

**Strengths:**

S1. The paper studies an interesting topic of OOD graph generalization.
S2. The paper presentation includes rich content, with tables and figures organized.
S3. The proposed method does not require ground-truth environment labels.

**Weaknesses:**

1. In Figure 1. (b), the false identified variant subgraph appears to be an expressiveness problem. There lack analysis from this perspective.
2. The method uses intra-class similarity, which seems intrinsically the same as [2] and impairs the novelty of the paper.
3. According to [3], learning invariant/spurious features without environment partition is fundamentally impossible if not given further inductive biases or information, which is also verified in [4]. Are there assumptions or additional information in this work to guarantee its environment inference?
4. As far as I know, GIL[1], which this work closely follows, has not released its code. The authors should provide code for this work to evidence the integrity of the experiments.
5. The authors mentioned that “In this paper, we use the results of baselines reported in Chen et al. (2022b); Li et al. (2022b) for comparisons”. This does not stand as a fait=r evaluation setting since different frameworks produce varying results.
6. In A.3, it’s mentioned that the authors “set the hyper-parameter α for C-VEI and baselines as 0.25, 0.5, 0.6, and 0.8 on SP-Motif, Graph-SST5, Graph-Twitter and DrugOOD”. Hyperparameters are typically searched in a range instead of given directly. Assigning the alpha value for each dataset seems like a leak of the test data information.
7. The paper writing is perfunctory. Algorithm 1 is by the name of “Pseudo code for the CIGA framework”, which looks like from another paper.

[1] Learning Invariant Graph Representations for Out-of-Distribution Generalization

[2] Learning causally invariant representations for out-of-distribution generalization on graphs

[3] ZIN: When and How to Learn Invariance Without Environment Partition?

[4] Rethinking invariant graph representation learning without environment partitions

**Questions:**

See weaknesses

---

### Official Review · Reviewer_hXrT · 2023-11-01

**Soundness:** 2 fair
**Presentation:** 3 good
**Contribution:** 2 fair
**Rating:** 5
**Confidence:** 3

**Summary:**

Motivated by the observation that class-wise spurious features are more likely shared by different classes despite high environment heterogeneity, this paper introduces a novel graph invariant learning framework, named C-VEI, which infers class-wise virtual environments with the assistance of labels and learning class-wise invariant features, instead of inferring underlying actual environments. The extensive experiments on several graph OOD benchmarks demonstrate the superiority of C-VEI.

**Strengths:**

1. This paper identifies several practical challenges in the underlying environment inference of previous methods: (a) the number of data from each environment can be too small to cover all classes; (b) the number of the underlying environments can be too large to infer reliably.
2. The CVE module is intuitive and believable, which can remove spurious correlations in the task setting of the paper.

**Weaknesses:**

1. The novelty of the proposed method may be doubtful. The similarity calculation in the CVE module seems redundant. Without the similarity calculation part, this method is not too different from supervised contrastive learning[1].
2. As mentioned above, the ablation study on the CVE module is not sufficient.

[1] Khosla, Prannay, et al. "Supervised contrastive learning." Advances in neural information processing systems 33 (2020): 18661-18673.

**Questions:**

1. After the similarity calculation part in the CVE module, samples from other classes are divided into “similar” samples and “dissimilar” samples, then an environment-based supervised contrastive loss is built. From the results, the loss stretches the distance between the samples in the anchor class and all samples in other classes. How could the environment division benefit the performance? Please provide some explanation and supplementary experiments.
2. Figure 4 is confusing. It seems the authors try to show that the proposed method can decouple the spurious correlation between the "House" class and the "Ladder" base. What about the spurious correlation between the "House" class and the "Tree" base and “Wheel” base? The figure is not intuitive enough.